# A Modified Actin (Gly65Val Substitution) Expressed in Cotton Disrupts Polymerization of Actin Filaments Leading to the Phenotype of Ligon Lintless-1 (*Li*_1_) Mutant

**DOI:** 10.3390/ijms22063000

**Published:** 2021-03-16

**Authors:** Yuefen Cao, Hui Huang, Yanjun Yu, Huaqin Dai, Huanfeng Hao, Hua Zhang, Yurong Jiang, Mingquan Ding, Feifei Li, Lili Tu, Zhaosheng Kong, Junkang Rong

**Affiliations:** 1The Key Laboratory for Quality Improvement of Agricultural Products of Zhejiang Province, School of Agriculture and Food Science, Zhejiang A&F University, Lin’an, Hangzhou 311300, Zhejiang, China; caoyaofen@126.com (Y.C.); 11816074@zju.edu.cn (H.D.); haoqiufeng87@126.com (H.H.); hzhang123@yahoo.com (H.Z.); yurongjiang746@126.com (Y.J.); 20110028@zafu.edu.cn (M.D.); lifei-fei@163.com (F.L.); 2National Key Laboratory of Crop Genetic Improvement, Huazhong Agricultural University, Wuhan 430000, Hubei, China; huangh93@126.com (H.H.); lilitu@mail.hzau.edu.cn (L.T.); 3State Key Laboratory of Plant Genomics, Institute of Microbiology, Chinese Academy of Sciences, Beijing 100000, China; yyj74@163.com

**Keywords:** cotton, Ligon lintless-1, gene mapping, transgenic cotton, F-actin

## Abstract

Dynamic remodeling of the actin cytoskeleton plays a central role in the elongation of cotton fibers, which are the most important natural fibers in the global textile industry. Here, a high-resolution mapping approach combined with comparative sequencing and a transgenic method revealed that a G65V substitution in the cotton actin Gh_D04G0865 (GhACT17D in the wild-type) is responsible for the *Gossypium hirsutum* Ligon lintless-1 (*Li*_1_) mutant (GhACT17DM). In the mutant GhACT17DM from *Li*_1_ plant, Gly65 is substituted with valine on the lip of the nucleotide-binding domain of GhACT17D, which probably affects the polymerization of F-actin. Over-expression of *GhACT17DM*, but not *GhACT17D*, driven by either a CaMV35 promoter or a fiber-specific promoter in cotton produced a *Li*_1_-like phenotype. Compared with the wild-type control, actin filaments in *Li*_1_ fibers showed higher growth and shrinkage rates, decreased filament skewness and parallelness, and increased filament density. Taken together, our results indicate that the incorporation of GhACT17DM during actin polymerization disrupts the establishment and dynamics of the actin cytoskeleton, resulting in defective fiber elongation and the overall dwarf and twisted phenotype of the *Li*_1_ mutant.

## 1. Introduction

Cotton is an important economic crop, providing most of the natural fibers used in the textile industry. High fiber yield and good quality are always the ultimate goals in the conventional breeding of upland cotton, which is a challenge for scientific researchers, because of the complex genetic basis of these plants. *Gossypium hirsutum* Ligon lintless-1 (*Li*_1_) is a fiber mutant [1], which has been used frequently to study the genetic mechanisms associated with fiber development in cotton. *Li*_1_ not only has seeds with very short lint fibers, but also displays other abnormal phenotypes such as twisted stems, curled leaves, and short plants, which have been confirmed to be caused by a dominant gene and result in a low level of survival in the homozygote [2,3]. The *Li*_1_ locus has been mapped to Chr.22 (D) [4,5,6], and the candidates for the *Li*_1_ gene vary among different studies [4,7,8,9] or lack supportive evidence of their roles in causing the mutation [10,11,12]. 

Actin proteins are required to construct the cell skeleton, maintain normal development, and undertake almost all activities in eukaryotic cells [13]. In higher plants, actin is encoded by a multigene family, whose members are involved in many aspects of plant growth and development. In poplars, the actin family consists of eight members that are differentially expressed in various tissues, but preferentially expressed in the stem phloem and xylem, suggesting that poplar actins are involved in wood formation [14]. In *Arabidopsis*, the actin gene family contains eight functional members. They are grouped into two ancient classes, a vegetative class that is expressed in leaves, stems, roots, petals, and sepals and a reproductive class that is expressed in pollen, ovules, and embryonic tissues [15,16,17]. The mis-expression of the *Arabidopsis ACT1* (*AtACT1*) iso-variant in vegetative tissues affects the dynamics of actin and actin-associated proteins, in turn, disrupting the organization of the actin cytoskeleton and normal development in *Arabidopsis* [18]. The actin cytoskeleton, through interaction with the ARP2/ARP3 complex, plays a crucial role in controlling the shape of several cell types, especially trichome cells in *Arabidopsis* [19,20]. *Arabidopsis ACT2* (*AtACT2*) functions in cell elongation and root formation, and an *AtACT2* dominant-negative mutation disturbs F-actin polymerization, causing aberrant cell morphology [21]. Another actin mutant (*AtACT8*) with a single nucleotide substitution also has dominant-negative effects on actin polymerization, resulting in functional defects of actin filaments, including frizzy filaments containing many kinks and severe curling and coiling-up of lateral shoots [22]. Taken together, the actin cytoskeleton is essential for cell elongation and tip growth and may be involved in the transportation of organelles and vesicles carrying membrane and cell wall components to the site of cell growth, as is the case in root hairs, trichomes, and fibers [23].

In cotton, 15 actin genes (*GhACT1-15*) have been cloned and characterized. Some evidence indicates that actin is important in maintaining the normal morphology of cotton fibers [23]. Upland cotton actin genes are differentially expressed in different tissues, and *GhACT1* is known to play a critical role in fiber elongation [23]. In other studies, a number of cytoskeletal genes have been found to be expressed specifically or preferentially in developing fiber cells. A comparative proteomic analysis between *Li*_1_ and its wild-type revealed that numerous cytoskeleton-related proteins are decreased remarkably in the fibers of *Li*_1_. Accordingly, the architecture of the actin cytoskeleton is severely deformed and the microtubule organization is moderately altered, while vesicle trafficking is dramatically disrupted [24]. Using sequencing-mapping and virus-induced gene silencing (VIGS), Thyssen et al. [11] and Sun et al. [12] recently indicated that the actin molecule encoded by *Gh_D04G0865* could be the best candidate for producing the *Li*_1_ mutant. Thyssen et al. [11] proposed “Poison subunit” model based on structural modeling of actin monomer and polymer, whereas Sun et al. [12] experimentally proved that model by in vitro polymerization of defective actin unit. However, there is no direct transgenic evidence available. More genetic and cellular evidence still needed to revealing the mechanism for F-actin polymerization in vivo.

We previously reported fine mapping of the *Li*_1_ locus in a gene-sparse region of about 0.3 cM and 1.2 Mb in length, where 36 genes have been annotated [10]. As candidate gene expression studies indicated that *Gh_D04G0865* is a promising candidate, we immediately constructed a larger F_2_ population to further fine map this gene and carried out transgenic verification of *Gh_D04G0865* in *Arabidopsis* and cotton. In the present work, the *Li*_1_ gene was further mapped to a narrower region, which co-segregated with *Gh_D04G0865*. Based on comparative sequencing, analysis of expression patterns, transgenic evidence, and filament array analysis, *Gh_D04G0865*, named *GhACT17D* in our experiments, mutated in *Li*_1_ plant. We named this mutant as *GhACT17DM,* the G65V substitution of which was found to affect F-actin elongation, causing the abnormal development of the *Li*_1_ mutant. 

## 2. Results

### 2.1. Fine Mapping and Sequence Analysis Indicated That GhACT17D Was a Promising Candidate Gene for the Li_1_ Mutation

Compared with the wild-type, the *Li*_1_ mutant not only exhibited short-lint on the seeds but also presented obvious differences in all morphological characteristics throughout its development (Figure 1D). The mutant demonstrated delayed growth with smaller and twisted stems, leaves, and flowers. Fewer and shorter trichomes were observed on the stems of the homozygote (*li_1_li_1_*), in which the plant height was less and seed setting lower than those of the heterozygote (*Li_1_li_1_*) (Appendix A). 

We had previously mapped the *Li*_1_ locus to a 0.3 cM region using two populations consisting of 1166 F_2_ plants, which was about 1.2 Mb in the published *G. raimondii* genome [10]. When the previously mapped markers around *Li*_1_ were aligned in the TM-1 reference sequence [25] (Figure 1B), most of the homologous loci of the single-strand conformation polymorphism (SSCP) markers were found, except for a few including P095, P257, and P191. They also appeared in the same order as before, except for P251 (Figure 1A,B). To further narrow the *Li*_1_ region, a total of 80 microsatellite markers (Appendix A) were selected from the TM1 genome sequence to screen for the polymorphic ones. After analyzing 2316 F_2_ plants, including 1111 used in our previous paper, 603 new F_2_ plants from the same cross combination and 602 F_2_ plants from E24-3583 × *Li*_1_, a genetic map containing 19 markers was constructed. Following this effort, *Li*_1_ was found to co-segregate with D04_1111, D04_1117, and D04_1118, which narrowed the search to a 0.04 cM region flanked by D04_1108 and D04_1124, corresponding to 392 kb of the TM-1 reference genome (Figure 1C, Appendix A). According to the gene annotation, this region contained eight genes, in which a beta-tubulin 7 (*Gh_D04G0862*) and an actin (*Gh_D04G0865*) gene (Figure 1C) were predicted to be two of the most-promising candidates to be the *Li*_1_ mutation (Jiang, et al., 2015). However, no expression of the beta-tubulin 7 gene was found in the stem and ovules of 0 DPA wild-type plants or *Li*_1_ mutants [10]. Therefore, we isolated the *Gh_D04G0865* sequences from the *Li*_1_ mutant and wild-type plants, and a single nucleotide substitution of G by T at the 194th base of the ORF from *Li*_1_ was detected (Figure 1E). According to these results, we deduced that *Gh_D04G0865* was the most-promising candidate gene causing the *Li*_1_ mutation and named it *GhACT17D*. The mutant from *Li*_1_ was named as *GhACT17DM.*

Using the SMART-RACE approach, the full-length cDNA of *GhACT17D* containing 1532 was cloned, and deposited in GenBank (accession numbers: MG132060). Sequence analysis revealed that the cDNA of *GhACT17D* contained a single ORF from nucleotides 63 to 1196, which was flanked by a 62-bp 5′-UTR and 336-bp 3′-UTR. The gene structure of *GhACT17D* showed that three introns split the coding region between the start and stop codons (Figure 1E). Furthermore, a 397-bp intron interrupted the 5′-UTR at -10 nucleotides upstream of the ATG initiation codon (Figure 1E). The splicing sites of all introns obeyed the GT-AG rule.

The *GhACT17D* ORF was 1134 bp in length and encoded a peptide containing 377 amino acid residues, comprising 38 strongly basic (+), 50 strongly acidic (−), 129 hydrophobic, and 86 polar amino acids. In a search against the NCBI protein sequence databases, the putative protein sequence of GhACT17D showed homology with actin proteins from other plants and contained the conserved nucleotide-binding domain (NBD) of the sugar kinase/HSP70/actin superfamily, which forms a deep cleft in which the nucleotide sits [26]. Eleven residues that compose this conserved feature were mapped to GhACT17D (Figure 2A). The GhACT17DM protein from the *Li*_1_ mutant had a single residue substitution (Gly to Val) at the 65th amino acid in the NBD region, which was the result of a single nucleotide mutation (G to T) at the 194th base in the ORF (Figure 1). 

### 2.2. Gly65 Is Highly Conserved and Crucial to GhACT17D

We investigated the sequence variation of *GhACT17D* in eight cultivars of five tetraploid cotton species. The coding regions of *GhACT17D* from these cultivars were sequenced, aligned, and used to construct a phylogenetic tree. All sequences were divided into two subgroups, A and D, and each group included eight cultivar sequences. *GhACT17D* sequences varied slightly either between different species or different cultivars, but no single nucleotide polymorphism (SNP) was found at the mutation site of *GhACT17DM* in the genome (Appendix A). We also analyzed the G65 position in GhACT17D of 36 cotton actin families with complete NBDs in the TM-1 genome [25] and eight *Arabidopsis* actin sequences, which showed that none of them varied in the position of G65 (Appendix A, Appendix A). These results indicated that the G65 in GhACT17D was highly conserved in paralogs and orthologs.

The physicochemical properties of the putative proteins showed that GhACT17D and GhACT17DM were stable, but the instability index and molecular weight of GhACT17DM were slightly higher than those of GhACT17D (Appendix A). The analysis of intolerant amino acid substitutions sorted by SIFT revealed that the change of G65 to V65 was deleterious (p = 0.00), which indicated that the GhACT17DM protein was altered phenotypically and functionally. The three-dimensional structure of the GhACT17D/GhACT17DM monomer was modeled with Swiss-Model Workspace using a major actin (SMTL id: 3ci5.1.A) with a Global Model Quality Estimation (GMQE) of 0.99 as the template, and G65 was found to be at the entrance of the NBD cleft (Figure 2B, labeled by box 1). When using another actin (SMTL id: 1nmd.1.A) with a GMQE of 0.98 as the template, a site for ligand SO2, in contact with R64, T204, T205, and A206, was found to be adjacent to the 65th amino acid (Figure 2C). Furthermore, the GhACT17D/DM homo-dimer was modeled with a cytoplasmic actin (SMTL id: 2oan.1.A) as the template (Figure 2D). These simulated protein structures indicated the importance of the position of G65 in GhACT17D. Compared to G65 in GhACT17D, the V65 in GhACT17DM emerged as a derivative of three branches (Figure 2C,D, labeled by box 1), which was attributed to the fact that valine is a non-polar branched-chain amino acid, while glycine is a small amino acid. The substitution of G65V would result in spatial change in the structure of the GhACT17DM molecule (Figure 2D), most likely affecting the polymerization of actin filaments.

### 2.3. Over-Expression of GhACT17DM in Transgenic Upland Cotton Results in the Li_1_ Mutant Phenotype 

Considering that actin is a constitutive cytoskeletal protein, the heterologous expression of *GhACT17DM* in *Arabidopsis* seemed insufficient to prove its function, although the over-expression of *GhACT17DM* resulted in more severely deformed morphology than the over-expression of *GhACT17D* (data not shown). Therefore, over-expression transgenic cotton lines were produced (Figure 3 and Appendix A). *GhACT17DM* sequences, containing an 1134-bp ORF along with a 48-bp 3′-UTR, were inserted into over-expression vectors driven by CaMV35s or GbEXPA2 promoters (pGbEXPA2), respectively. CaMV35s is a constitutive promoter and pGbEXPA2 is the preferential promoter for quickly elongating cotton fibers [27]. Twenty-five independent transgenic over-expression lines of pGbEXPA2::*GhACT17DM* were obtained and all the lines generated short fibers, without obvious changes in plant morphology. After screening by Southern blotting, the transgenic lines with low copy numbers (EM4, EM10, and EM24) were selected for further analysis (Appendix A). With *GhACT17D*-specific primers, the *GhACT17DM* expression levels were found to be significantly increased in fibers (10 DPA) of EM4, EM10, and EM24, compared to the wild-type and nulls (Appendix A). The expression level was closely related to the mature fiber lengths of the transgenic lines and the fibers were shorter when the expression level was higher (Figure 3A, Appendix A). Only two CaMV35s::*GhACT17DM* transgenic lines (3M3-1 and 3M3-6) were obtained, because the constitutive strong expression of *GhACT17DM* affected somatic embryogenesis, which resulted in the same phenotype as the *Li* mutant, with dwarf plants, curly leaves, and much shorter fibers (Figure 3A,C,D). Similar to the result obtained from the pGbEXPA2::*GhACT17DM* transgenic lines, the *GhACT17DM* transcript levels of 3M3-6 were significantly higher than those of 3M3-1, and the leaves of 3M3-6 were more heavily crinkled than those of 3M3-1.

In order to further confirm that *GhACT17DM* over-expression, and not that of *GhACT17D*, resulted in the mutant phenotype, transgenic lines (3W7 and 3W46) over-expressing *GhACT17D* driven by CaMV35 were also obtained. There was no obvious phenotype difference, although the expression levels were much higher than those in the wild-type (Figure 3C, Appendix A). 

### 2.4. An Abnormal F-actin Structure Results in Defective Growth and Development of the Li_1_ Mutant

To gain further insight into the cellular mechanisms underlying the severely dysfunctional growth phenotype of the *Li*_1_ mutant, we performed live cell imaging of the actin cytoskeleton in both elongating cotton fibers at 2 DPA and expanding leaf epidermal cells. Remarkably, F-actin architectures in both the *Li*_1_ fiber cells and leaf epidermal cells had shorter F-actin fragments, along with long F-actin cables, compared with the wild-type control. The defective F-actin organization in the *Li*_1_ mutant resembled that of the dominant-negative *Arabidopsis* act2-2D mutant, which displays actin polymerization defects [21]. Moreover, we monitored and measured the parameters of the F-actin dynamics in rapidly elongating fiber cells at 2 DPA. Both the growth and shrinkage rates of F-actin in *Li*_1_ fibers were about twice those of the wild-type control (Figure 4 and Appendix A). The F-actin density was significantly higher in the *Li*_1_ fibers than the wild-type control. On the contrary, the skewness and parallelness of F-actin decreased in the *Li*_1_ fibers. Taken together, these results indicated that actin polymerization may be disturbed in the *Li*_1_ mutant, resulting in defective F-actin assembly with more short fragments offering less stability.

## 3. Discussion

### 3.1. Dominant-Negative Mutation of the Gh_D04G0865 Gene Causes the Li_1_ Phenotype 

The *Li*_1_ mutant has attracted extensive attention since its discovery in 1929 [1] and much effort has been put into cloning the gene responsible [2,4]. Many possible candidate genes and related pathways have been suggested, but most of the results, especially those from gene expression analyses, vary considerably [28,29,30]. In 2015, we proposed that *Gh_D04G0865* (numbered as P258) was one of the most promising candidate *Li*_1_ mutant genes, based on fine genetic mapping and gene expression analysis [10]. Thyssen et al. also identified *Gh_D04G0865* as the *Li*_1_ candidate, labeled by a SNP marker, CFB11927 (23,783,515 bp in *G. hirsutum* Chr D04), using sequencing-mapping and VIGS [11]. The experiment carried out by Thyssen et al. used a 354-bp fragment from the conservative domain of actin C-terminal sharing high similarity among actin members that would not discriminate the actin families in VIGS, which resulted in a *Li*_1_-like phenotype but in a different mechanism [12]. Benefiting from the whole genome sequencing of upland cotton [25], we now know that 37 actin genes with complete NBDs sharing high identity occur in the TM-1 genome, including *Gh_D04G0865* (Appendix A, labeled by black square). Therefore, Sun et al. used a 3′UTR of *Gh_D04G0865* for VIGS to show that it caused little change in wild-type seedlings and partially recovered the curled leaves to be normal in *Li*_1_ seedlings after specifically inhibiting *Gh_D04G0865* expression [12]. Sun et al. also discovered that the G65V substitution in Gh_D04G0865 disturbed filament polymerization in vitro but failed to generate gene-overexpression cotton.

In this study, the *Li*_1_ locus was further narrowed to a region of 392 kb, in which only eight genes exist. According to nucleotide variation and gene expression studies, *Gh_D04G0865* was believed to be the most promising candidate for the *Li*_1_ mutation. Further stable transgenic validation confirmed that the overexpression of *GhACT17DM*, driven by either the CaMV35 or the pGbEXPA2 promoter in cotton caused the *Li*_1_-like phenotype. Therefore, our results directly support the conclusion that *Gh_D04G0865* is the mutant gene associated with *Li*_1_ and its G65V substitution results in the abnormal phenotype of the *Li*_1_ mutant.

### 3.2. Incorporation of the Mutated Version of GhACT17DM Impaired Actin Polymerization and Cotton Growth with a Dosage-Related Effect

Actin is one of the most conserved proteins in eukaryotes. Structural predictions suggested that the G65V substitution would affect the polymerization capability of GhACT17DM, as confirmed by previous in vitro assays [12]. Importantly, our live-cell observations revealed that *Li*_1_ cells showed defective dynamic behavior of the actin cytoskeleton, compared with that of the wild-type control (Figure 4). Actin is one of the most abundant proteins in cells and numerous actin encoding genes show redundant roles. Thus, knocking out the function of a single actin gene usually gives rise to no obvious change in phenotype. However, a point mutant in an important domain of the actin molecule that often produces a dominant-negative effect. It should be noted that F-actin networks could still form in *Li*_1_ cells, because GhACT17DM only accounts for a small portion of actin proteins in the abundant actin pool. The incorporation of the mutated version of GhACT17DM impaired actin polymerization, thus generating a defective actin cytoskeleton with short F-actin fragments in *Li*_1_ cells (Figure 4 and Appendix A). Integrating more defective actin variants resulted in more severe defects of the actin cytoskeleton, showing a dosage effect. Accordingly, the homozygous *Li*_1_ presents a more twisted and dwarf phenotype than the heterozygote and does not often survive, which causes a lower number of mutant plants than expected in the F_2_ offspring [3,6,10]. In our transgenic cotton lines, the higher the *GhACT17DM* expression, the shorter the mature fiber length or more crinkled the plant. These results indicated that more copies of *GhACT17DM*, or more transcripts, made plants more severely abnormal, which revealed the positive dosage effect of *GhACT17DM*.

## 4. Materials and Methods

### 4.1. Plant Materials and Fine Genetic Mapping

A total of 12 varieties of five tetraploid cotton species, *G. barbadense*, *G. darwinii*, *G. hirsutum*, *G. mastelinum*, and *G. tomentosum*, were used in this study (Appendix A) to produce mapping populations and transgenic plants for sequence analysis. *Li*_1_ is a *G. hirsutum* Ligon-lintless-1 mutant provided by Dr. Xiongming Du from the Cotton Research Institute of China. Three mapping populations were developed by crossing the *Li*_1_ mutant with *G. barbadense* var. Hai7124 and E24-3583 (Totally 2316 F_2_ plants, including 1111 plants from Hai7124 × *Li*_1_ in 2012, 603 new F_2_ plants from the same cross combination in 2016 and 602 F_2_ plants from E24-3583 × *Li*_1_). *G. hirsutum* acc. YZ1 was used to produce transgenic cotton. The plants were grown in a greenhouse at Huazhong Agricultural University in Wuhan using standard farm management practices, according to relevant approvals for biotechnology research. The other four tetraploid cotton species were used to study gene expression, sequence similarity, and copy number.

The generation of ABD2-GFP marker lines was produced using G. hirsutum (accession R15) as the recipient of the 35S::ABD2-GFP construct [31]. Crosses were made between *Li*_1_ mutants and ABD2-GFP marker lines. In the segregated F_2_ progeny, *Li*_1_ homozygous lines and wild-type lines expressing the ABD2-GFP marker were used for live cell imaging, and subsequent comparative analysis on F-actin dynamics was performed between *Li*_1_ fibers and the wild-type control. Images of elongating fibers were obtained from cotton fibers at 2 days-post-anthesis (DPA), and images of leaf epidermal cells were obtained from the first true leaf of 2-week-old cotton seedlings.

Genomic DNA was extracted from fresh leaves using the CTAB method [32], which was slightly modified, as described by Ding [7]. DNA from the F_2_ populations was used to map *Li*_1_. In order to narrow the map of *Li*_1_, 19 simple sequence repeat (SSR) markers were selected from those reported in the targeted region [25], based on a previously published map [10] (Appendix A). The bordering SSRs were used to screen the recombinant F_2_ plants. Further, SSR markers were applied to determine the location of the recombinant events and finally to narrow the location of the *Li*_1_ mutation.

### 4.2. Gene Cloning and Sequence Analysis

The DNA extracted from different genotypes and species was used to amplify the candidate gene for sequence analysis. Total RNA was extracted from fresh young leaves with a TianGen RNAprep pure kit (Tiangen, China). The full-length cDNA was obtained using SMART-Rapid Amplification of cDNA Ends (RACE) approaches (SMARTer^®^ RACE 5′/3′ Kit, Clontech, USA). The gene-specific primers (Appendix A) for 5′ and 3′ RACE were designed from the coding regions. Approximately 2 μg of high-quality total RNA was used for the 5′ and 3′ SMART-RACE protocols, strictly following the manufacturer’s instructions. The PCR products obtained from the cDNA and genomic DNA amplification were ligated into the pMD 18-T vector (Takara, China) and transformed into *E. coli* cells (DH5α), before sequencing at BGI (Shenzheng, China).

### 4.3. cDNA Preparation and RT-PCR

To conduct reverse transcription (RT)-PCR analysis, approximately 1 μg of total RNA from each sample was used to synthesize the first-strand cDNAs in a 20-μL reaction volume using Moloney-Murine Leukemia Virus reverse transcriptase (Takara) with Oligo d(T)18 primer. The synthesized cDNAs were used as templates in the following PCR reactions.

For each target gene, PCR amplification was performed using Taq polymerase (Takara) with gene-specific primer pairs (Appendix A). The PCR was conducted in a thermal cycler (Bio-RAD, Singapore), using the following procedure: pre-denaturation at 94 °C for 3 min, followed by 30 cycles of 30 s at 94 °C, and 60 s at 68 °C, for each gene. The cotton gene *ChUB7* (DQ116411) was used as an internal control [33].

### 4.4. Quantitative Real-Time PCR (qRT-PCR)

Quantitative PCR was carried out to analyze *GhACT17D/GhACT17DM* expression in the *Li*_1_ mutant and wild-type, the primers were listed in Appendix A. The cDNAs used to detect gene expression were the same as those used in the RT-PCR analysis. The reaction was conducted on a BioRad CFX connect™ real-time PCR System (BioRad, USA) using Takara Taq^TM^ (Takara) with EvaGreen (Biotium), according to the manufacturer’s instructions. The amplification of the target gene was monitored with an EvaGreen fluorescence signal on every cycle. The relative expression level was standardized to the expression level of *ChUB7* (DQ116411) (Appendix A). The cycle threshold value (Cq) was used as a measure of the starting copy number of the target gene, and gene relative expression level =1/2^(CqGene-CqUbq7), error value = S/sqrt(3), S=standard deviation [33].

Quantitative PCR was carried out to analyze gene expression in the transgenic cotton. Flower buds were marked on the day of anthesis (0 DPA), 10 DPA bolls were harvested, hulls were stripped, and fibers and ovules were frozen immediately in liquid nitrogen and stored at −70 °C until use. Total RNA was extracted using the thiocyanate method and 3 µg of total RNA were reverse-transcribed to cDNA using SuperScript II reverse transcriptase (Invitrogen, USA). Quantitative RT-PCR was performed as previously described [34] on a 7500 Real-Time PCR system (Applied Biosystems).

### 4.5. Phylogenetic Analysis

The sequence data used here were retrieved from GenBank databases using BLASTN/P programs (http://blast.ncbi.nlm.nih.gov (accessed on 1 February 2021)) on the National Center for Biotechnology Information (NCBI) database. Multiple alignments of the nucleotide and deduced amino acid sequences were performed using ClustalW [35], and the results were manually adjusted with GeneDoc 2.7 [36]. A phylogenetic tree was constructed by the neighbor-joining method with 1000 bootstrap replications, using MEGA 5 [37]. The ExPASy ProtPara program (http://web.expasy.org/protparam/ (accessed on 1 February 2021)) was used to analyze the physicochemical properties. The SIFT tool was used to predict the effects of amino acid substitutions on proteins, with probabilities of < 0.05 considered deleterious [38]. The Swiss-Model Workplace was used to carry out protein structure homology modeling [39].

### 4.6. Vector Construction and Genetic Transformation in Cotton

To obtain an over-expression construct, an 1134-bp ORF fragment, along with the 48-bp 3′-untranslated region (UTR) of *GhACT17DM* and *GhACT17D* was amplified by the primers with attB1 and attB2 adaptors and ligated into the over-expression vectors pGWB408 (driven by the CaMV35s promoter) and pGWB407-GbEXPA2 (driven by the PGbEXPA2 promoter) using Gateway technology [40].

*Agrobacterium tumefaciens* (EHA105) was used to infect the hypocotyls of *G. hirsutum* YZ1, according to methods described previously [41,42].

### 4.7. Southern Blotting in Transgenic Cotton

The genomic DNA was extracted using a Plant Genome Extracting kit (TIANGEN, China) and 20 µg DNA were digested completely with *Hind*Ⅲ, separated by 0.8% gel electrophoresis, transferred into a nylon membrane (Amersham, UK), and hybridized with digoxin-labeled DNA fragments of *NPTⅡ* at 42 °C overnight. The membrane was incubated with anti-AP in 20 mL blocking solution at 37 °C for 40 min and washed three times. The CSPD was added and signaling was detected in a darkroom. Detail procedures were as described in the DIG High Prime DNA Labelling and Detection Starter Kit II (Roche, Switzerland).

### 4.8. Analyzing the Dynamic Behavior of F-actin

Live-cell imaging and video were carried out under a spinning disk confocal microscope (UltraView VoX, Perkin Elmer) equipped with the Yokogawa Nipkow CSU-X1 spinning disk scanner, the Hamamatsu EMCCD 9100-13 and the Nikon TiE inverted microscope containing the Perfect Focus System. Acquired images were analyzed using Volocity (Perkin Elmer), ImageJ (http://rsbweb.nih.gov/ij (accessed on 1 February 2021)), as described previously [31]. The F-actin severing frequency was defined as the number of severing events per 100 μm^2^ per second. The elongation and shrinkage rates (μm/s) were measured as previously described [43,44]. F-actin skewness and density (%) were measured according to an established approach [45]. Parallelness, which was defined as the mean angular difference, was applied to evaluate the F-actin orientations as described previously [46].

## 5. Conclusions

In conclusion, our findings revealed that the G65V substitution in GhACT17DM is responsible for the *Li*_1_ mutation, and incorporation of GhACT17DM during actin polymerization disturbs the establishment and dynamics of the actin cytoskeleton, resulting in defective fiber elongation and the overall dwarf and twisted phenotype of the *Li*_1_ mutant.

## Figures and Tables

**Figure 1 ijms-22-03000-f001:**
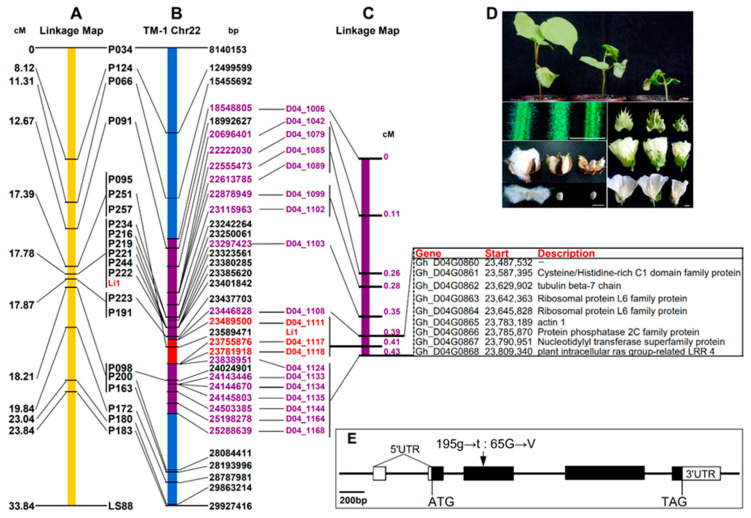
Genetic and physical maps of the *Li*_1_ mutant and the candidate genes. (**A**), The fine map of the *Li*_1_ gene from a previous paper (Jiang et al., 2015). (**B**), The physical map of the *Li*_1_ gene, based on the published TM-1 reference genome (Zhang, et al., 2015). (**C**), The genetic map of *Li*_1_ and the candidate genes constructed using all segregating data from current research. The *lines* connect the same DNA sequence in different maps. (**D**), The varied phenotypes from different genotypes of *Li*_1_. Scale Bars, 1 cm. (**E**), Gene structure of GhACT17D. Boxes represent exons; filled parts of boxes represent the coding region. The position of the point mutation in GhACT17DM is indicated by a black arrow.

**Figure 2 ijms-22-03000-f002:**
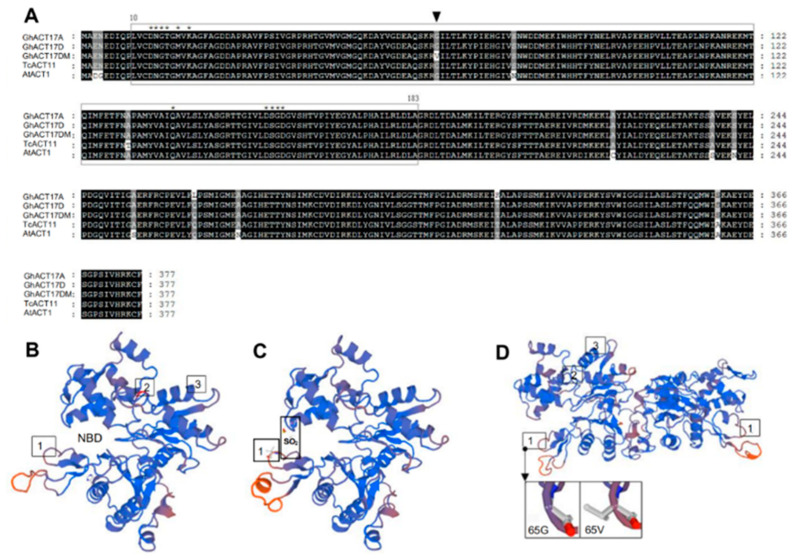
The multiple alignment and protein structure homology model of GhACT17D/GhACT17DM. (**A**), The multiple alignment of GhACT17D/DM homologs and the other actin proteins. The residents labeled by asterisks (*) show the conserved features of the sugar kinase/HSP70/actin superfamily nucleotide binding domain (NBD) model. The NBD in GhACT17D/DM is boxed in (amino acids 10–183). Val, the point of mutation in GhACT17DM, is labeled by a black triangle. GhACT17A is a homolog of GhACT17D listed in Appendix A. TcACT11 and AtACT1 are actin genes from other plant (Appendix A). (**B**), GhACT17D monomer model using 3ci5.1.A as the template. (**C**), GhACT17DM monomer model using 1nmd.1.A as the template. The V65 in GhACT17DM emerged as a derivative of three branches. (**D**), GhACT17D/DM homo-dimer model using 2oan.1.A as the template. The arrow refers to the enlarged view of the difference between G65 in GhACT17D and V65 in GhACT17DM. 1, 65th amino acid; 2, 265th amino acid; 3, 320th amino acid.

**Figure 3 ijms-22-03000-f003:**
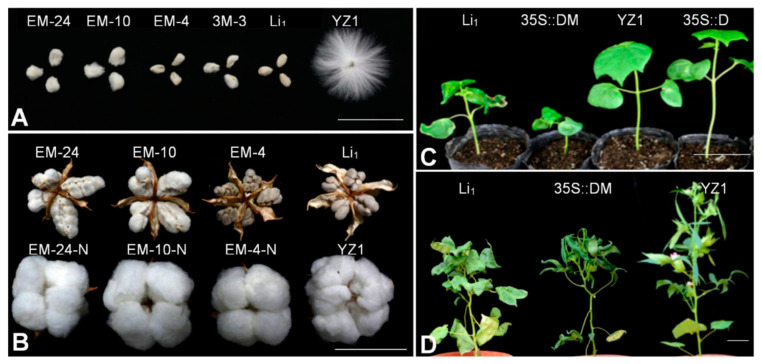
Phenotype of transgenic cotton. (**A**), The mature fibers of transgenic and wild-type cotton. (**B**), The mature bolls of transgenic cotton and null. (**C**,**D**), Phenotype of *GhACT17DM* transgenic cotton shows dwarf plant and crinkled leaves. 35S:DM, overexpression of GhACT17DM with CaMV35; 35S:D, overexpression of GhACT17D with CaMV35; EM24/10/4-N, the normal plant segregated from transgenic cotton. Scale bars, 5 cm.

**Figure 4 ijms-22-03000-f004:**
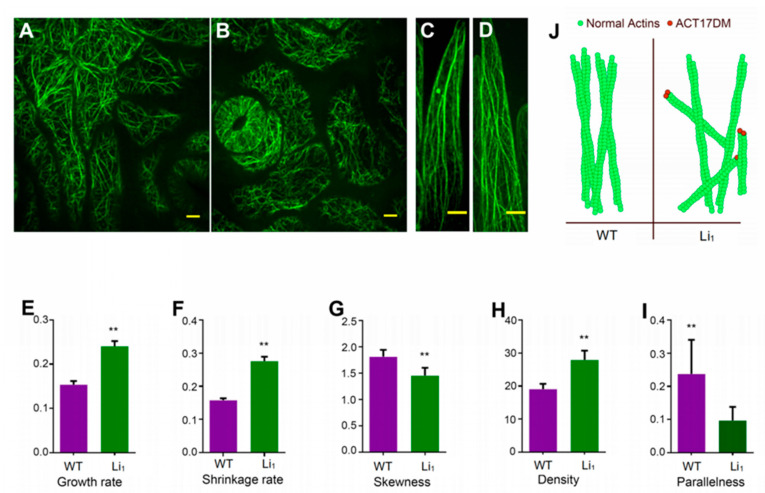
The architectures and dynamic behaviors of F-actin in the *Li*_1_ mutant and the wild-type**.** (**A**), F-actin architectures in epidermal pavement cells of the wild-type. (**B**), F-actin architectures in epidermal pavement cells of the *Li*_1_ mutant. (**C**), F-actin architectures in fiber cells of the wild-type. (**D**), F-actin architectures in fiber cells of the *Li*_1_ mutant. (**E**–**I**), Comparison of the parameters (growth rate, shrinkage rate, skewness, density, and parallelness) of F-actin dynamics in fiber cells between the *Li*_1_ mutant and the wild-type. (**J**), Proposed model II for interpreting the effects of GhACT17DM in F-actin elongation. Black arrows represent the elongation direction. Scale bars, 10 µm. The two asterisks represent highly significant difference.

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
