# Peer review of "A Modified Actin (Gly65Val Substitution) Expressed in Cotton Disrupts Polymerization of Actin Filaments Leading to the Phenotype of Ligon Lintless-1 (Li1) Mutant"

_ijms, 2021, doi:10.3390/ijms22063000_

Round 1
Reviewer 1 Report
Using the map-based strategy, Cao et al. cloned the dominant Li1 mutation in cotton causing poor fiber quality and other growth defects. The candidate was functionally validated through transgenic complementation test, which provided a direct evidence for the identity of Li1. Based on the F-actin structure analysis, the authors speculated that the abnormal growth phenotype of the Li1 mutant might be trigged by dysfunctional actin cytoskeleton. I appreciate the nice work performed by Cao et al, and feel sorry for the novelty was clouded by the precedent publications. Although Thyssen et al. and Sun et al. failed to present the experimental data using homologous or heterologous systems to support the G65V substitution is the causative mutation, their VIGS data is indicative also. I would like to suggest Cao et to change their tone in the manuscript. I understand that being scooped is a nightmare, but almost most scientists experience at least once. Overall, this is an fantastic job.
Below are some minor points need to be considered:
- Lines 75-76. The authors’ claim will cause unnecessary dispute. I suggest to delete “Gh_D04G0865 has not been proven to be the gene causing the Li1 mutation”. Just simply point out that the direct transgenic evidences are missing in their work.
- Line 67, suggest to change the use of italics in “genes”
- Line 234, “Li-1” or “Li1 “ ? Please keep formatting consistent.
- Line 242, please insert “the” to read “…but most of the results…”
- Line 243, suggest to remove “the” to read “…(numbered as P258) was one of the most promising…”
Author Response
Thank you for your affirmation of our work, we will make persistent efforts. Here, we answered and revised the questions you pointed out in the paper one by one.
- Lines 75-76. The authors’ claim will cause unnecessary dispute. I suggest to delete “Gh_D04G0865 has not been proven to be the gene causing the Li1 mutation”. Just simply point out that the direct transgenic evidences are missing in their work.
A: The sentence “Gh_D04G0865 has not been proven to be the gene causing the Li1 mutation” has been deleted. Thanks for your very friendly suggestion.
- Line 67, suggest to change the use of italics in “genes”
A: Revised in the paper.
- Line 234, “Li-1” or “Li1 “? Please keep formatting consistent.
A: it is “Li1”. Revised.
- Line 242, please insert “the” to read “…but most of the results…”.
A: Revised in the paper.
- Line 243, suggest to remove “the” to read “…(numbered as P258) was one of the most promising…”.
A: Revised in the paper.
Thanks again.
Reviewer 2 Report
The manuscript written by Cao et al presents some interesting findings about Lini1 mutation and its impact on cotton lint characteristics.
However, the reader is easily confused about which mutation is actually described (GhACT17DM or GhACT17D) and is the one that is linked to the deformed phenotypes. Also the mutation GhACT17A is not extensively described. Some results that are given as supplementary data should be given in the manuscript (please see the attached file).
Overall althought the results are interesting, they are presented in a confusing and misleading manner so the re-organization and reforming of the structure of the results and discussion section is necessary to improve the quality of the manuscript.

Author Response
Thank you very much for your useful comments and suggestions on our paper. In order to make the article clearer, we decide to remove the work on GhACT17A from the manuscript because it was not related to the phenotype of Li1. GhACT17A would appear only as a cotton actin gene in the article and is listed supplementary 5. GhACT17DM is the mutant of GhACT17D (Gh_D04G0865). Here, we answered and revised the questions you pointed out in the paper one by one.
- Lines 84-85.
A: Revised. Now it is “Gh_D04G0865, named GhACT17D in our experiments, mutated in Li1 plant. We named this mutant as GhACT17DM, the G65V substitution of which was found to affect F-actin elongation, causing the abnormal development of the Li1 mutant. ”
- Lines 115-121.
A: Sorry for “Arabidopsis thaliana subgenome”, which was a slip of the pen, it should be “Gossypium hirsutum subgenome”. Here, we have deleted all the experiment data about GhACT17A through this paper. Supplemental Fig. 1 would be removed.
- Lines 154-158.
A: GhACT17D is a normal gene from wild type cotton, and GhACT17DM is a mutant gene from Li1 plants. We have added an explain— “The mutant from Li1 was named as GhACT17DM.” —at Lines 114-115, and deleted the analysis on GhACT17A. Data of supplemental table 6 related to GhACT17A also has been deleted.
- Line 191.
A: The style of “insufficient” has been revised, thanks.
- Line 335.
A: It has been changed to “Quantitative PCR was carried out to analyze GhACT17D/GhACT17DM expression in the Li1 mutant and wild-type, the primers were listed in supplemental table 3.”
Reviewer 3 Report
Authors of submitted manuscript experimentally proved through transgenic work that Gly65Val substitution in an actin gene disrupts polymerization of actin filaments resulting in Li1 phenotype. The manuscript should be available to public after revisions.
- Title. The manuscript will benefit if author consider to change the title with emphases on novelty of this manuscript, which is ectopic expression of modified actin gene (Gly65Val substitution) in cotton and Arabidopsis disrupts polymerization of actin filaments which lead to Li1 phenotype. Two previous independent studies mapped the Li1 gene to the same locus as this study. Such consistency is good thing, but that is not the novelty which will attract the reader attention.
- Introduction and discussion. Authors cited two previous papers, including Thyssen et al. [11] and Sun et al. [12]; however, did not mention important information. Particularly, Thyssen et al. [11] proposed “Poison subunit” model based on structural modeling of actin monomer and polymer, whereas Sun et al. [12] experimentally proved that model by in vitro polymerization of defective actin unit (authors mentioned). Evidently authors achieved the same result by using stable cotton and Arabidopsis transformation, which confirmed the “Poison subunit” model previously proposed by Thyssen et al. [11].
Another point in discussion should be clarified: Thyssen et al. used 354 bp from the C-terminal domain of actin protein; considering very high sequence similarity among actin members the VIGS will not discriminate and suppress the family of genes. Despite different mechanism it produced the same effect – less actin protein in Li1 mutant (confirmed by Western blot) and in VIGS plants (same phenotype). Sun et al. were unable to reproduce Li1 phenotype targeting only one actin unit by using a 3′UTR for VIGS.
- Methods:
- Plant materials: What amount of plants in each mapping population? Please specify which ABD2-GFP marker lines were used for crossings.
- qRT-PCR: please provide reference gene and calculation method.
- Imaging: please provide instrument and parameters used for images in this work
- Figure 4. Images can be improved. Differences in actin cytoskeleton are not clear between wild type and Li1 developing fibers. Authors may try to capture a single fiber cell and different developing time.
- Supplemental data 1: typo – “phynotype”. The authors provided the mapping data internally used in the lab without explaining how to read the data.
Author Response
Dear reviewer:
We have read the comments and suggestions from you seriously and uploaded the revised manuscript. Following is a point-by-point response to your comments. If you have any question, please feel free to ask. I will be happy to answer and complete.
Best regards.
Junkang Rong
- Question: The manuscript will benefit if author consider to change the title with emphases on novelty of this manuscript, which is ectopic expression of modified actin gene (Gly65Val substitution) in cotton and Arabidopsis disrupts polymerization of actin filaments which lead to Li1 phenotype. Two previous independent studies mapped the Li1 gene to the same locus as this study. Such consistency is good thing, but that is not the novelty which will attract the reader attention.
Answer: very good suggestion, we have changed the title to be “A modified actin (Gly65Val substitution) expressed in cotton disrupts polymerization of actin filaments leading to the phenotype of Ligon lintless-1 (Li1) mutant”.
- Question: Introduction and discussion. Authors cited two previous papers, including Thyssen et al. [11] and Sun et al. [12]; however, did not mention important information. Particularly, Thyssen et al. [11] proposed “Poison subunit” model based on structural modeling of actin monomer and polymer, whereas Sun et al. [12] experimentally proved that model by in vitro polymerization of defective actin unit (authors mentioned). Evidently authors achieved the same result by using stable cotton and Arabidopsis transformation, which confirmed the “Poison subunit” model previously proposed by Thyssen et al. [11].
Another point in discussion should be clarified: Thyssen et al. used 354 bp from the C-terminal domain of actin protein; considering very high sequence similarity among actin members the VIGS will not discriminate and suppress the family of genes. Despite different mechanism it produced the same effect – less actin protein in Li1 mutant (confirmed by Western blot) and in VIGS plants (same phenotype). Sun et al. were unable to reproduce Li1 phenotype targeting only one actin unit by using a 3′UTR for VIGS.
Answer: Also very good suggestion, we have added some information about two important previous papers in our introduction and discussion. Please see Line 76-79 and Line 252-259.
Sun et al have confirmed the “Poison subunit” proposed by Thyssen et al via in vitro polymerization. More genetic and cellular evidence still needed to revealing the mechanism for F-actin polymerization in vivo. Combined fine genetic mapping and live-cell imaging, we identified the G65V of GhACT17DM is able to affect the F-actin elongation and lead to abnormal development of Li1 mutant.
Thyssen et al using a 354-bp fragment in VIGS may result in a Li1-like phenotype in a different mechanism. And the 3’UTR for VIGS is also insufficient to produce Li1 phenotype. By using stable transgenic cotton, we confirmed the G65V of GhACT17DM disrupted the F-actin organization in vivo.
3.1 Question: Plant materials: What amount of plants in each mapping population? Please specify which ABD2-GFP marker lines were used for crossings.
Answer: The plant number in each mapping population (mentioned in Result Line 105) have been added in Line 296-297 of Methods . ABD2-GFP marker line’s information has been added in Line 302-303.
3.2 Question: qRT-PCR: please provide reference gene and calculation method.
Answer: the reference gene and calculation method were added in Line 342-346.
3.3 Question: Imaging: please provide instrument and parameters used for images in this work
Answer: The instrument and parameters used in this research for analyzing the dynamic behavior of F-actin have been added in Line 381-385.
3.4 Question: Figure 4. Images can be improved. Differences in actin cytoskeleton are not clear between wild type and Li1 developing fibers. Authors may try to capture a single fiber cell and different developing time.
Answer: Thank you for your good advice, but this is the best level we can achieve at present. We will improve it in the future, hopefully.
3.5 Question: Supplemental data 1: typo –“phynotype”. The authors provided the mapping data internally used in the lab without explaining how to read the data.
Answer: The notes— “ the Arabic numeral “5” in “phenotype” represents Li1’s phenotype; “3” represents wide type. The Arabic numeral “1” in genotype of each marker represents Li1’s genotype; “3” represents wide type; “2” represents heterozygote. ” —have been added in Line 427-429 of this paper.
Round 2
Reviewer 2 Report
Although some corrections were done and now its more clear that GhACT17D refers to the gene/protein and the GhACT17DM refers to the name of the mutated line, it should be consistent throughout the text in order to be finally accepted for publication.
Author Response
We have revised the problems you pointed out. Items related to GhACT17D or GhACT17DM have been modified to be consistent throughout the text. Please, see the revised version at Line 21/22/28/131/158/163/169/173/176/181/378/379. Many thanks for your suggestions. I really appreciate it.